# Tailoring lipid management interventions to reduce inequalities in cardiovascular disease risk management in primary care for deprived communities in Northern England: a mixed-methods intervention development protocol

Yu Fu ,[1,2] Eugene YH Tang,[1] Sarah Sowden ,[1] Julia L Newton,[1,3] Paula Whitty[2,4]

For numbered affiliations see end of article.

**Correspondence to**
Dr Yu Fu;
yu.fu@newcastle.ac.uk

## ABSTRACT

**Introduction** Hyperlipidaemia contributes a significant proportion of modifiable cardiovascular disease (CVD) risk, which is a condition that disproportionally affects disadvantaged socioeconomic communities, with death rates in the most deprived areas being four times higher than those in the least deprived. With the national CVD Prevention programme being delivered to minimise risk factors, no evidence is available on what has been implemented in primary care for deprived populations. This study describes the protocol for the development of a tailored intervention aiming to optimise lipid management in primary care settings to help reduce inequalities in CVD risks and improve outcomes in deprived communities.

**Methods and analysis** A mixed-methods approach will be employed consisting of four work packages: (1) rapid review and logic model; (2) assessment and comparison of CVD risk management for deprived with non-deprived populations in Northern England to England overall; (3) interviews with health professionals; and (4) intervention development. A systematic search and narrative synthesis will be undertaken to identify evidence-based interventions and targeted outcomes in deprived areas. General practice-level data will be assessed to establish the profile of lipid management, compared with the regional and national levels. Health professionals involved in the organisation and delivery of routine lipid management to deprived populations will be interviewed to understand the implementation and delivery of current lipid management and associated challenges. The prototype intervention will be informed by the evidence generated from workpackages 1–3, which will be reviewed and assessed using the nominal group technique to reach consensus. Training and skills development materials will also be developed as needed.

**Ethics and dissemination** Ethics approval has been obtained from the Faculty of Medical Sciences Research Ethics Committee at Newcastle University, UK. Findings will be disseminated to the participating sites, participants, commissioners, and in peer-reviewed journals and academic conferences.

### STRENGTHS AND LIMITATIONS OF THIS STUDY

⇒ This study will develop a tailored lipid management intervention for deprived populations to help reduce health inequalities, using multiple methods.
⇒ Multiple data sources will be used to assess and compare cardiovascular disease risk management for deprived with non-deprived populations in Northern England to England overall.
⇒ Primary care staff needs and challenges in delivering current lipid management and resources related to implementation will be identified.
⇒ Some limitations to the study design include exclusion of non-English studies, publication bias, quality of data and selection bias in the rapid evidence review.

## INTRODUCTION

Hyperlipidaemia contributes a significant proportion of modifiable cardiovascular disease (CVD) risk, which is the leading cause of mortality and morbidity in England and Europe,[1 2] accounting for a third of deaths in the UK. Hyperlipidaemia, a high level of cholesterol or triglycerides in the blood, can be inherited, is often found in people who are overweight, have alcohol abuse or have an unhealthy diet.[3] Elevated levels of blood lipids represent a major risk factor for the development of coronary heart disease and other cerebrovascular diseases including stroke, transient ischaemic attack (TIA) and peripheral arterial disease. People with a history of these events are also at increased risk of experiencing subsequent CVDs. Management of CVDs also places a significant economic burden on the National Health Service (NHS), with an estimated cost of £7.4 billion per annum.[4]

Both national and international guidelines recommend the use of statins in people at risk of CVD.[1 5] Their use aims to reduce the synthesis of cholesterol, but evidence suggests that there is an underuse of lipid-lowering drugs among eligible patients.[6 7] CVD is also a condition that is strongly associated with health inequalities and disproportionally affects disadvantaged socioeconomic communities. People in disadvantaged socioeconomic groups experience a higher prevalence of CVD events but poorer outcomes and premature mortality, leading to the fact that people in the most deprived areas in England are four times more likely to die prematurely than those in the least deprived.[8] In addition, those living in socioeconomically disadvantaged neighbourhoods are found to have poor engagement with preventive health services, even though they are likely to benefit from screening and early treatment.[9] This may lead to an exacerbation of existing health inequalities. The National Institute for Health and Care Excellence (NICE) guidance has recognised socioeconomic status as an additional factor that contributes to CVD risk. The NHS Long Term Plan[10] has also identified CVD as a clinical priority and stressed the wider impact on health inequalities, highlighting that heart disease-related mortality is the single largest contributor to the life expectancy gap between the most and least deprived. However, it failed to establish how health inequalities could be approached or addressed within local systems.

The North East of England is consistently ranked as having the highest poverty levels and the lowest health outcomes in England.[11] Scotland has established a programme to support general practices caring for the most deprived communities (the 'Deep End' project)[12] and, in early 2020, local General practitioners (GPs), Public Health leaders and academics collaborated to form a Deep End Steering Group for the North East and North Cumbria (NENC). Funding was then granted from the North East and North Cumbria Integrated Care System (NENC ICS) Prevention strand to establish and codesign a Deep End network for the region. The Deep End NENC network consists of 35 practices; practices identified as Deep End are those that fall into the 10% most deprived practice populations in England. These practices have between 95.7% and 57.7% of registered patients living in the most deprived 15% of indices of multiple deprivation data zones. Due to the high rates of long-term conditions, unhealthy diets and physical inactivity, together with other competing priorities,[13] people in areas of deprivation are likely to face greater challenges in managing

CVDs. Ongoing effects from the pandemic are exacerbating these challenges and include difficulties attending review appointments in person, digital poverty impeding remote review, low levels of health literacy resulting in misunderstandings about the need to continue long-term treatments, and closure of other support services,[14] potentially widening health inequalities.

The NHS has set up the national CVD Prevention programme[15] which aims to develop targeted interventions to minimise risk factors by maximising diagnosis and treatment, accompanied by the GP contract to commission a new national CVD prevention audit for primary care.[16] However, no evidence is available on what and how the intervention has been implemented and for what health outcomes for deprived populations. There is therefore an urgent need to seek a theoretical underpinning to tailor the national programme in this context, which could support the CVD element of the NHS post-COVID-19 recovery plan with the region.

This study will examine the literature and practice-level data and undertake engagement with staff who provide primary care for deprived populations to define the components and mechanisms through which lipid management can be optimised to meet the identified needs. The study aims to (1) synthesise the evidence on interventions for deprived populations with CVDs or those with high risks and understand the outcomes associated with these interventions, (2) assess CVD risk management for deprived populations in the NENC in comparison with non-deprived populations in Northern England and with England overall in order to identify clinical gaps and needs, (3) investigate the implementation and delivery of current interventions for patients with CVDs and those with high risks, and (4) tailor and optimise the national prevention programme to suit the context and needs of deprived communities.

## METHODS AND ANALYSIS
### Study design
A mixed-methods approach will be employed to inform the development of the intervention comprising a rapid review, a population-based observational study and qualitative interviews. Four work packages (WPs) are proposed (figure 1).

A project advisory group consisting of 6–8 members will be established to involve key opinion leaders across core fields, who will advise at each project stage, review intervention components for the consensus process and help disseminate the study outputs. Members will recruit from the Deep End network, ICS Prevention Board, Academic Health Science Network (AHSN) NENC lipid steering group, regional professional leads for lipid management, public members and academics with methodology expertise. This group will meet quarterly with the research team to oversee the execution of the study and provide advice and assistance.

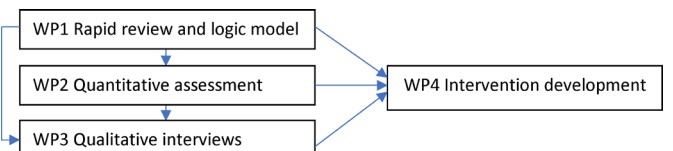

**Figure 1** Study design and related WPs. WPs, work packages.

## Patient and public involvement

Patients' experiences are central to the research question and outcomes, although the focus of this project is on clinicians' experiences. The Deep End Steering group, consisting of local GPs, representatives from the North East Commissioning Support Unit (NECS), Newcastle University Medical School, Health Education England North East and the Postgraduate School of Primary Care, Directors of Public Health, NHS England, the Northern Cancer Alliance and local voluntary, community and social enterprise organisations, was consulted in order to shape the research focus and question, methods of data collection and dissemination. While the focus of the research is initially on clinicians' experience, the development of a patient and public involvement strategy was recognised in the consultation as an urgent requirement, and this is currently being developed. The results of this study will be widely shared via the Public Involvement and Community Engagement network for the National Institute for Health Research (NIHR) Applied Research Collaboration NENC.

## WP1: rapid review and logic model (months 1–3)

WP 1 will be a rapid review and synthesis of current evidence, aiming to identify evidence-based interventions for lipid management in deprived areas and targeted outcomes. The review will be conducted following Cochrane guidance on rapid reviews.[17] A logic model will be developed, informed by existing literature to describe how lipid management works in theory to benefit services and patients.

### Type of studies

Empirical studies (ie, original data collection) describing the setting, problem addressed, resource requirements, aim, intervention components, provider, method of delivery and objective and subjective outcomes will be included if conducted in an Organisation for Economic Co-operation and Development (OECD) country[18] (to ensure a degree of commonality in health system and socioeconomic and demographic context), published in peer-reviewed scientific journals, within the last 10 years (to mirror the NHS long-term plan) and in the English language.

### Type of participants

Studies that focus on people with disadvantaged socioeconomic status (education, income, occupation, social class, deprivation, poverty or an area-based proxy for deprivation derived from place of residence) will be included. Adults with CVD including angina, previous myocardial infarction, revascularisation, stroke or TIA or symptomatic peripheral arterial disease, and those who do not have established CVD but are identified as having a high risk of developing CVDs[1] considering age, ethnicity, socioeconomic status, body mass index, history of taking antihypertensive or lipid modification therapy, Cardiovascular Risk Score (QRISK) ≥10%, diabetes, nephropathy, familial hypercholesterolaemia or other inherited disorder of lipid metabolism, and other underlying medical conditions or treatment including people treated for HIV, with serious mental health problems, taking medicines that can cause dyslipidaemia or with autoimmune disorders.

### Type of interventions

Multifaceted interventions delivered to deprived populations that aim to optimise care by maximising diagnosis and/or treatment to minimise individual risk factors will be considered.

### Type of outcome measures

Studies with individual, area-based or both types of measures of socioeconomic deprivation will be included. This may be measured according to several characteristics including income, employment, education, disability, crime, housing and services and living environment deprivation. Because there is no universal recommendation for core outcome sets in studies on CVD prevention,[19–21] studies will be eligible for inclusion regardless of outcomes measured or reported for health outcomes, this may include vascular-related outcomes, cognitive and functional outcomes, lifestyle, medical risk factors, cardioprotective medications and patient-reported outcome measures. Any measures of professionals', patients' and/or families' knowledge, attitudes or satisfaction will also be included.

### Study identification

Cochrane CENTRAL, MEDLINE, PsycInfo, CINAHL will be searched for eligible studies. Detailed search strategies will be developed for each database. A preliminary search strategy developed for MEDLINE is designed by YF and validated by an information specialist (online supplemental material 1). This search strategy was piloted in MEDLINE on 16 October 2021.

### Study selection

Identified citations will be exported to Endnote X9[22] for deduplication and screening. A random selection (10%) of study titles and abstracts will be screened independently by another researcher. Full text will be retrieved where citations appeared to meet the eligibility criteria or where a decision to exclude will not be made on the information provided. Any discrepancies will be resolved by discussion with a third researcher.

### Data extraction

Data will be extracted on author's first name, publication date, location (country in which the study was undertaken), study design, sample size, intervention details, control/comparison groups (if any), outcome measures and results, using a data extraction sheet that will be piloted on two retrieved study reports. Accuracy and consistency will be monitored through random double extraction of 10% included studies by an independent researcher. Any discrepancies will be resolved by discussion with a third

researcher. Where a study appears to have multiple citations, original authors will be contacted for clarification. All information from multiple citations will be used if no replies received.

## Quality assessment

Quality appraisal of included studies will be performed using standardised tools adapted for purpose. Appropriate Critical Appraisal Skills Programme tool will be used according to the study design, a random sample (10%) will be independently assessed by another researcher. Any discrepancies will be resolved by discussion with a third researcher.

## Data synthesis

A narrative synthesis will be undertaken following Popay et al's[23] approach to conducting synthesis systematically and transparently. It will focus on the intervention components, effects of the interventions and mechanisms leading to the outcomes. Studies, interventions and outcomes will be examined and grouped according to the aim and components of the interventions. The variation on different characteristics of health systems will be taken into account when interpreting the intervention across OECD countries. A logic model will be produced to present context, intervention components

and outcomes. Possible unintended adverse outcomes will also be provided.

## WP2: assessment and comparison of CVD risk management for deprived with non-deprived populations to England overall (months 2–4)

WP2 will be a population-based observational study comparing retrospective data from practices in deprived communities in the NENC, practices in regional non-deprived communities and national practice-level data obtained from publicly accessible datasets and anonymised data requested from the NECS that securely house primary and secondary care datasets.

## Data sources

The primary data source for this study will be the GP Practice Profiles[24] via Fingertips, a publicly accessible web tool containing national general practice profiles generated for all Quality and Outcomes Framework (QOF)[25] 2019/2020 with a list size of at least 750 patients. Available practice-level data include local demography, QOF domains and patient satisfaction. Other data sources used include the QOF, OpenPerscribing[26] and data requested from the NECS via Secondary Uses Services (SUS)[27] data (table 1).

**Table 1** Data sources and variables

| Data source | Description | Level of data available | Variables and variable description |
|---|---|---|---|
| GP Practice Profiles | Date reported by GPs to the NHS that refers to all patients in a practice | ▶ Individual practice | ▶ Practice size<br>▶ Mean practice age<br>▶ Deprivation score<br>▶ Age groups<br>▶ Percentage of patients positive experiences as 'good'<br>▶ Percentage of practice access rated by patients as 'good'<br>▶ Percentage with a long-term condition<br>▶ Education status<br>▶ Working status<br>▶ Life expectancy by sex |
| QOF | An indication of the overall achievement of a practice through a points system, concerning clinical, public health, public health—additional services, and quality improvement. It also has cardiovascular group data. | ▶ Individual practice | ▶ QOF score<br>▶ Total on the AF register, prevalence<br>▶ Total on the CVD-primary prevention (CVD-PP) register, prevalence<br>▶ Total on the CHD register, prevalence<br>▶ Total on the HF register, prevalence<br>▶ Total on the LVSD register, prevalence<br>▶ Total on the HYP register, prevalence<br>▶ Total on the PAD register, prevalence<br>▶ Total on the STIA register, prevalence |
| Open Prescribing | Imports national prescribing data published by NHS Business Services Authority | ▶ Individual practice<br>▶ CCG level | ▶ Total statin<br>▶ Total low and medium intensity statin |

AF, atrial fibrillation; CCG, Clinical commissioning group; CHD, coronary heart disease; CVD, cardiovascular disease; HF, heart failure; HYP, hypertension; LVSD, left ventricular systolic dysfunction; NHS, National Health Service; PAD, peripheral arterial disease; QOF, Quality and Outcomes Framework; STIA, stroke and transient ischaemic attack.

## Study population

The study population are patients aged 16 and above who have registered with the 34 Deep End practices in the NENC and have been diagnosed with any form of CVDs recorded on the QOF from 2019 to 2020. The study comparators are the patients registered in non-Deep End practices in the region and all registered patients in England where data are available. Data are aggregated to the GP practice level in which variables will be summarised if they are at the patient level.

## Data analysis

Descriptive analysis will be used to provide a quick and low cost approach to assess CVD risk management and give descriptive statistics or investigate relationships between factors. GP practice code will be used to link data across all datasets. Due to the nature of the aggregated data available from the public sources used (Fingertips[24] and QOF), it will be not possible to control any of the comparisons for age, gender, deprivation or ethnicity. Descriptive statistics, using means, SD and range, will be used to compare the practice profile of the 34 Deep End practices with non-Deep End in the region and England average level. The prevalence of risk factors and statin prescribing will be analysed with an appropriate statistical test (ie, two-sample t-test, single sample t-test and paired t-test), which will yield p values that indicate the statistical significance of any differences between Deep End, non-Deep End and England level. A paired t-test will be used to understand whether there was a difference in outcomes before and at the time of the COVID-19 pandemic. CIs for differences in means, medians or percentages will be calculated. All significance tests will be performed at the 5% level. Stata V.16 will be used to facilitate data analysis.

## WP3: interviews with health professionals (months 3–8)

WP3 will be qualitative interviews with staff involved in the organisation and delivery of routine lipid management in practices that are part of the Deep End Network. The aim of the interviews will be to understand the implementation and delivery of current lipid management and identify their needs and challenges.

## Participants and recruitment

All health professionals involved in the management of CVDs in the practice are eligible to take part including GPs, pharmacists, assistant practitioners, practice nurses and social prescribers.

Study recruitment will be supported by the Deep End practice Network, who will send an email containing brief study information to healthcare professionals working in participating practices. Health professionals can express their interests by responding to the email straight to the research team. A reminder will be sent to those who have not responded in 2 weeks. Maximum variation sampling will be used to ensure a broad representation of health professionals on dimensions including job titles/roles, grade, specialty, length of working and demographics.

Reasons given by practices for declining to participate will be recorded to inform feasibility assessment to further studies.

## Data collection

With participants' informed written consent, semistructured interviews will be conducted via telephone or online (ie, Zoom or MS Teams) for up to 60 mins. A topic guide was drafted to address the research questions and piloted with two primary care health professionals to ensure the questions prepared are relevant for the context and acceptable. Questions considered important but not originally included were also sought from the pilot interviews, and the topic guide was amended accordingly (online supplemental material 2). As interviews continue, the topic guide will also allow a deeper exploration of emerging themes and participants' feedback, while maintaining a consistent core of questions. Data collection will end when data saturation is reached indicating no new information is discovered.

## Data analysis

Interviews will be digitally recorded (with consent), transcribed and data managed using NVivo V.12, a qualitative software programme to assist with the organisation and coding of data. Data will be analysed using Framework Analysis, which provides a systematic approach to sifting, charting and sorting material using the key themes and issues. Initial line-by-line coding will be undertaken. The connections and relationships of these codes will be explored, contributing to the development of themes. An analytical framework will have been developed as the coding process progresses and themes emerge. Codes and themes from each transcript will be compared and integrated using the constant comparison process, enabling continuous updates on the interview topic guide and the thorough interpretation of the study data. To ensure trustworthiness and rigour of the analysis, the coding framework will be developed and assured by double coding of a random sample of transcripts (10%) as a validity check and exploring alternative interpretations of the data.

## WP4: intervention development (months 7–9)

WP4 will develop the prototype of the intervention in collaboration with the Academic Health Science Network NENC which delivers the CVD Prevention programme, part of which includes a national programme mandated by NHS England and NHS Improvement.

Guided by the Medical Research Council Framework for developing and evaluating complex interventions,[28] the development of intervention will be informed by integrating the outcomes of the literature evidence, current CVD management profile and stakeholder engagement undertaken in WPs1–3, in an iterative and progressive approach. The national programme and its key components will be examined against the gaps, needs and

challenges identified to consider the wider context. The prototype intervention will be designed taking account of health professionals' existing commitments in these practices and challenging working environments. Training and skills development materials for health professionals will also be developed to facilitate them in delivering the tailored intervention. The logic model produced in WP1 will be refined to map key intervention processes and outcomes.

## Design

The prototype intervention will be reviewed and assessed by the project advisory group, guided by a nominal group technique, a consensus method that allows for the generation of views and thoughts from group participants while maintaining anonymity throughout.[29]

## Data collection

The group will be provided with details of the interventions and refined logic model, to seek further comments and explore if the intervention is feasible, acceptable and implementable in the context. The APEASE criteria[30] will be used to determine the acceptability, practicability, effectiveness, affordability, side effects and equity aspects of the intervention. The nominal group technique will involve two main sections:

1. The group will be asked to provide their comments on the intervention, training materials and logic model. All comments will be collated and grouped into main themes for each member to rate their top 10 priorities of the comments. Group ratings will be summated, and the group's collective top 10 priorities will be presented to the group and discussed.
2. Each will rerate the group's top 10 priorities and provide a weighting for their top 10 comments in the scale ranging from 1=least important to 100=most important. These weightings will be summated after the meeting, which will be used to refine the intervention and the logic model. The refined version will be sent out to each member for further comments. It is expected that this process will be repeated twice until a census is reached.

## Data analysis

The initial listing of comments, clarification and discussion of comments in section 1 of the nominal group technique (listed above) will be analysed thematically, with further discussion with the research team. The scale data generated in section 2 of the nominal group technique will be averaged so that the comments can be reordered according to weighted ranked priority. The individual/group rankings produced in sections 1 and 2 will be compared to estimate the level of agreement between the sections and to observe the process of reaching consensus. First, these comparisons will be made by calculating the percentage agreement between the sections, in terms of the comments that appear in the top 10 priorities

each time. Second, the movement in ranking between sections 1 and 2 will be estimated using Cohen's kappa statistic of chance-corrected agreement.[31] A kappa value of >0.40 is considered to represent a moderate level of agreement.[32]

## ETHICS AND DISSEMINATION

Ethics approval has been sought from the Faculty of Medical Sciences Research Ethics Committee at Newcastle University (reference no.: 2209/14251) UK, a letter of support has also been issued by the NENC Deep End Network.

Dissemination will be led by the research team and supported by the project advisory group. Reports will be produced and shared with the NENC ICS Prevention Board, Inequalities Board, Deep End network, National Institute for Health Research (NIHR) Applied Research Collaboration (ARC) NENC and AHSN NENC. The findings will be disseminated to the participating sites, participants, commissioners and in peer-reviewed journals and academic conferences.

**Author affiliations**
[1]Faculty of Medical Sciences, Population Health Sciences Institute, Newcastle University, Newcastle upon Tyne, UK
[2]NIHR Applied Research Collaborative North East and North Cumbria, Cumbria, Northumberland and Tyne and Wear NHS Foundation Trust, Newcastle upto Tyne, UK
[3]Academic Health Science Network for the North East and North Cumbria, Newcastle upon Tyne, UK
[4]North East Quality Observatory Service, Cumbria Northumberland Tyne and Wear NHS Foundation Trust, Newcastle upon Tyne, UK

**Acknowledgements** We thank NENC Deep End Steering Group and NECSU for their support in this study.

**Contributors** YF, JLN and PW conceived the study design. YF drafted and revised the manuscript and obtained ethics approval. EYHT, SS, JLN and PW independently reviewed and contributed to revising and approving the final version.

**Funding** This project is supported by the National Institute of Health Research (NIHR) [Applied Research Collaboration North East and North Cumbria (NIHR200173)]. The views expressed are those of the author(s) and not necessarily those of the NIHR or the Department of Health and Social Care. EYHT (National Institute for Health Research (NIHR) Clinical Lecturer) is funded by the NIHR. The views expressed in this publication are those of the author(s) and not necessarily those of the NIHR, NHS or the UK Department of Health and Social Care.

**Competing interests** None declared.

**Patient and public involvement** Patients and/or the public were involved in the design, or conduct, or reporting or dissemination plans of this research. Refer to the Methods section for further details.

**Patient consent for publication** Not applicable.

**Provenance and peer review** Not commissioned; externally peer reviewed.

**ORCID iDs**
Yu Fu http://orcid.org/0000-0003-4972-0626
Sarah Sowden http://orcid.org/0000-0001-9359-3463

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
