## [Reviewer comments · BMJ Open]

ARTICLE DETAILS

TITLE (PROVISIONAL)	Tailoring lipid management interventions to reduce inequalities in cardiovascular disease risk management in primary care for deprived communities in Northern England: a mixed methods intervention development protocol
AUTHORS	Fu, Yu; Tang, Eugene; Sowden, Sarah; Newton, Julia L.; Whitty, Paula

VERSION 1 – REVIEW

REVIEWER	Natasha Curran University College London Medical School, Anaesthesia and Pain Management
REVIEW RETURNED	20-Feb-2022

GENERAL COMMENTS	The protocol could be improved by close attention to style and strengthening the section on PPI and being tighter on the language around the qualitative analysis. Specific comments: p3 Line 9, English does not make it clear if CVD or hyperlipidaemia is the condition which has the risk. Line 30 Add 'General' to 'practice level'.. as not mentioned previously p4 Line 10 - Describe 'Deep End' practices as ?X...general practices, confusing to reader who has not heard of this, not explained until later in protocol. p8 line 18. some of the English not clear and poor grammar e.g. what are you trying to say in this sentence? "However, patients' experiences are central to the research question and outcomes, which has been recognised as an urgent agenda to aim to develop a patient and public involvement strategy." p11 line 28- What is "CHD register"? p12 line 32 I don't understand this: "It provides right time access to A&E" p13 line 26 Difficult to understand what the second half of this sentence means "Pharmacists and social prescribers in general practice also have a key role in supporting the management of dyslipidaemia, therefore consulting clinician and the facilitator a healthcare assistant would also be invited." line 40 What is the relevance of COVID-19 exposure? line 48 Language around the qualitative research loose. What will the pilot with ' a couple of professionals' do/change? How will rigour be assured with the topic guide being continuously updated in line with emerging themes and participants' feedback? For sure needs to iterate and explore some themes more deeply but need to ensure that still focuses on need of the study. Suggest removing 'approximately' - from in front of 20-25 interviews, you've already
--

	said 'anticipate'.
--	--------------------

REVIEWER	Chan Huak Biostatistics Unit, Yong Loo Lin School of Medicine, National University Health System
REVIEW RETURNED	23-Mar-2022

GENERAL COMMENTS	Thank you for detailed proposal. My comments. the study is going to be performed on the NE of England only? if that so, there will be no comparator on the findings. Possible to tag an estimate timeline for each WPs? WP1. ok WP2. The t-test (or Mann-Whitney U test) will be used to compare continuous variables and the χ^2 test (or Fisher's exact test) will be used for similar comparisons of categorical variables. which groups are being compared here? As stated, if there is no comparator data from a least-deprived area, the results will not be useful. WP3. The professionals to be sampled for the interviews are rather diverse (doctors, pharmacists, healthcare assistant, nurses, etc), how to know that data saturation has reached after 20-25 interviews?
---

VERSION 1 – AUTHOR RESPONSE

Reviewer 1: Dr. Natasha Curran, University College London Medical School	Responses
The protocol could be improved by close attention to style and strengthening the section on PPI and being tighter on the language around the qualitative analysis.	The PPI section has been revised in the Methods and Analysis section. The qualitative analysis section has been revised and more information on how this will be carried out has been provided in WP3.
p3 Line 9, English does not make it clear if CVD or hyperlipidaemia is the condition which has the risk.	This has been revised in the Abstract.
Line 30 Add 'General' to 'practice level'.. as not mentioned previously	This has been added as suggested in the Abstract.
p4 Line 10 - Describe 'Deep End' practices as ?X....general practices, confusing to reader who has not heard of this, not explained until later in protocol.	This has been revised in the Strengths and limitations section, extra information has also been added to the Introduction section of this study.

p8 line 18. some of the English not clear and poor grammar e.g. what are you trying to say in this sentence? "However, patients' experiences are central to the research question and outcomes, which has been recognised as an urgent agenda to aim to develop a patient and public involvement strategy."	The Patient and public involvement section has been revised.
p11 line 28- What is "CHD register"?	This has been revised in Table 1.
p12 line 32 I don't understand this: "It provides right time access to A&E"	This has been revised in Table 1.
p13 line 26 Difficult to understand what the second half of this sentence means "Pharmacists and social prescribers in general practice also have a key role in supporting the management of dyslipidaemia, therefore consulting clinician and the facilitator a healthcare assistant would also be invited."	This has been revised and simplified in the Methods and Analysis section for WP3. "All health care professionals involved in the management of CVDs in the practice are eligible to take part including GPs, pharmacists, assistant practitioners, practice nurses and social prescribers."
line 40 What is the relevance of COVID-19 exposure?	This has been taken out.
line 48 Language around the qualitative research loose. What will the pilot with ' a couple of professionals' do/change? How will rigour be assured with the topic guide being continuously updated in line with emerging themes and participants' feedback? For sure needs to iterate and explore some themes more deeply but need to ensure that still focuses on need of the study. Suggest removing 'approximately' - from in front of 20-25 interviews, you've already said 'anticipate'.	This has been revised in the data collection and data analysis sections for the WP3, in the Methods and Analysis section. "A topic guide that has already been developed will be piloted with a couple of health professionals prior to data collection to ensure the questions prepared are relevant for the context and acceptable. Questions considered important but not included will also be sought from pilot interviews. As interviews continue, the topic guide will also be updated in line with emerging themes and participants' feedback." "Codes and themes from each transcript will be compared and integrated using the constant comparison process, enabling continuous

	updates on the interview topic guide and the thorough interpretation of the study data.”
--	--

Reviewer 2 Dr. Chan Huak, Biostatistics Unit, Yong Loo Lin School of Medicine, National University Health System	Responses
Thank you for detailed proposal.	Thank you.
The study is going to be performed on the NE of England only? if that so, there will be no comparator on the findings.	The qualitative study will be undertaken in Northern England (North East and North Cumbria) only; however, publicly available data will also be obtained for England. WP2 is a population based observational study comparing retrospective data from general practices in deprived communities in the North East and North Cumbria, with practices in non-deprived communities in the region and with national practice-level data. The text in the Methods and Analysis for WP2 has been revised to make this clearer.
Possible to tag an estimate timeline for each WPs?	Timeline has been added to each WP. WP1 (months 1-3) WP2 (months 2-4) WP3 (months 3-8) WP4 (months 7-9)
WP1. ok	Thank you.
WP2. The t-test (or Mann-Whitney U test) will be used to compare continuous variables and the χ^2 test (or Fisher’s exact test) will be used for similar comparisons of categorical variables. which groups are being compared here? As stated, if there is no comparator data from a least-deprived area, the results will not be useful.	WP2 is a population based observational study comparing retrospective data from practices in deprived communities in the NENC, practices in regional non-deprived communities and national practice-level data. The text in the Methods and Analysis for WP2 has been revised to make this clearer.
WP3. The professionals to be sampled for the interviews are rather diverse (doctors, pharmacists, healthcare assistant, nurses, etc), how to know that data saturation has reached after 20-25 interviews?	Thanks for your comment. We will employ a maximum variation sampling to ensure a broad representation of health care professionals in various primary care settings.

	Qualitative data collection will be guided by data saturation. We have taken out the anticipated sample size.
--	---

VERSION 2 – REVIEW

REVIEWER	Natasha Curran University College London Medical School, Anaesthesia and Pain Management
REVIEW RETURNED	13-Apr-2022

GENERAL COMMENTS	Dear Yu and colleagues. Thank you for comprehensively addressing the comments from the original reviews. At the bottom of page 11 - MRC should be written out.
---

REVIEWER	Chan Huak Biostatistics Unit, Yong Loo Lin School of Medicine, National University Health System
REVIEW RETURNED	12-Apr-2022

GENERAL COMMENTS	Thank you for making the extensive changes. 1 minor comment. WP2. pg 10. lines 18-35 will the comparison of the Deep End practices with non-Deep End in the region and England average level to be adjusted for relevant confounders, eg, demo... stated in line 32, the above comparisons will be over time, can elaborate on this and what stats analysis will be performed?
--

VERSION 2 – AUTHOR RESPONSE

Reviewer 1: Dr. Natasha Curran, University College London Medical School	Responses
Dear Yu and colleagues. Thank you for comprehensively addressing the comments from the original reviews.	Thank you.
At the bottom of page 11 - MRC should be written out.	This has been revised.

Reviewer 2 Dr. Chan Huak, Biostatistics Unit, Yong Loo Lin School of Medicine, National University Health System	Responses
---	------------------

Thank you for making the extensive changes.	Thank you.
WP2. pg 10. lines 18-35 will the comparison of the Deep End practices with non-Deep End in the region and England average level to be adjusted for relevant confounders, eg, demo...	This has been clarified that it will be not possible to control any of the comparisons for age, gender, deprivation or ethnicity due to the nature of the aggregated data available from the public sources used (Fingertips and QOF).
stated in line 32, the above comparisons will be over time, can you elaborate on this and what stats analysis will be performed?	This has been edited. A paired t-test will be used to understand whether there was a difference in outcomes before and at the time of the COVID-19 pandemic.

VERSION 3 – REVIEW

REVIEWER	Chan Huak Biostatistics Unit, Yong Loo Lin School of Medicine, National University Health System
REVIEW RETURNED	31-May-2022
GENERAL COMMENTS	Thank you